# Effects of White Fish Meal Replaced by Low-Quality Brown Fish Meal with Compound Additives on Growth Performance and Intestinal Health of Juvenile American Eel (*Anguilla rostrata*)

**DOI:** 10.3390/ani13182873

**Published:** 2023-09-09

**Authors:** Wenqi Lu, Haixia Yu, Ying Liang, Shaowei Zhai

**Affiliations:** Engineering Research Center of the Modern Industry Technology for Eel, Ministry of Education, Fisheries College of Jimei University, Xiamen 361021, China; wenqilu2022@163.com (W.L.); haixiayu2319@foxmail.com (H.Y.); lingyingnanping@jmu.edu.cn (Y.L.)

**Keywords:** low-quality brown fish meal, compound additives, growth performance, antioxidant potential, intestinal health, *Anguilla rostrata*

## Abstract

**Simple Summary:**

There is a high demand for white fish meal in the diet of eels (*Anguilla* spp.). A reduced supply and higher price of white fish meal heavily limit the sustainability of the eel aquaculture industry. It is essential to explore a practical strategy to reduce the consumption of white fish meal. In the current study, we investigated the effects of low-quality brown fish meal with compound additives to replace the high-quality white fish meal on the parameters of growth and intestinal health status of juvenile American eels (*Anguilla rostrata*). The results implied that low-quality brown fish meal combined with compound additives could successfully replace 20% white fish meal without adversely affecting the intestinal health status of this fish species.

**Abstract:**

With a reduced supply and increased price of white fish meal (WFM), the exploration of a practical strategy to replace WFM is urgent for sustainable eel culture. A 70-day feeding trial was conducted to evaluate the effects of replacing WFM with low-quality brown fish meal (LQBFM) with compound additives (CAs) on the growth performance and intestinal health of juvenile American eels (*Anguilla rostrata*). The 300 fish (11.02 ± 0.02 g/fish) were randomly distributed in triplicate to four groups (control group, LQBFM20+CAs group, LQBFM30+CAs group and LQBFM40+CAs group). They were fed the diets with LQBFM replacing WFM at 0, 20%, 30% and 40%, respectively. The CAs were a mixture of *Macleaya cordata* extract, grape seed proanthocyanidins and compound acidifiers; its level in the diets of the trial groups was 0.50%. No significant differences were found in the growth performance between the control and LQBFM20+CAs groups (*p* > 0.05), whereas those values were significantly decreased in LQBFM30+CAs and LQBFM40+CAs groups (*p* < 0.05). Compared to the control group, the activity of glutamic-pyruvic transaminase was significantly increased in LQBFM30+CAs and LQBFM40+CAs groups, while lysozyme activity and complement 3 level were significantly decreased in those two groups (*p* < 0.05). There were decreased antioxidant potential and intestinal morphological indexes in the LQBFM30+CAs and LQBFM40+CAs groups, and no significant differences in those parameters were observed between the control group and LQBFM20+CAs group (*p* > 0.05). The intestinal microbiota at the phylum level or genus level was beneficially regulated in the LQBFM20+CAs group; similar results were not shown in the LQBFM40+CAs group. In conclusion, with 0.50% CA supplementation in the diet, LQBFM could replace 20% of WFM without detrimental effects on the growth and intestinal health of juvenile American eels and replacing 30% and 40%WFM with LQBFM might exert negative effects on this fish species.

## 1. Introduction

Eels (*Anguilla* spp.) belong to the fish species with high-quality and rich nutrients and have had a reputation as “ginseng in water” since ancient times. As one of the most important aquatic products with high prices in international trade, eels make important contributions to the world fisheries economy [1]. As a carnivorous fish species, the protein requirement of eels is relatively high, and fish meal of the highest quality is the predominant source of protein. Because of the difference in fish sources, fish meal is usually divided into white fish meal (WFM) and brown fish meal (BFM). WFM is mainly manufactured from fish with white muscle including hake, cod, blue whiting, etc., and the main raw material of BFM is fish with red muscle including anchovy, jack mackerel, sardines, etc. Traditionally, high-quality WFM is used as the primary source of feed protein due to its high protein content, balanced amino acid composition, rich nutrients, high freshness and good digestibility, and the percentage of this WFM in the formula feed of eels is about 60–70% [2]. However, the availability of high-quality WFM is limited due to the downward trend in global fish meal production caused by El Nino events and the overexploitation of wild stocks. Moreover, the increasing demand for aquaculture production exacerbates the shortage of this fish meal [3,4]. Attributed to the ever-increasing cost of WFM, the feed cost continues to be a major challenge for the eel industry. Exploration into an inexpensive protein source that can replace the high-quality WFM in eel feed is highly needed.

Despite all the efforts in the research of WFM replacements from plant protein [5,6], husbandry animal protein [7,8] and single-cell protein [9] that has been carried out, the replacement effect and ratio are unsatisfactory due to the antinutritional factors present, nutrient deficiency, poor palatability, etc. Compared to other proteins, BFM has a similar nutritional profile to WFM, but the price varies greatly depending on its quality. It seems that relatively low-quality BFM (LQBFM) can be the realistically practical option to replace the high-quality WFM in eel feeds. There are many products of BFM with similar nutrients as high-quality WFM, but they are classified into the lower grade due to their high levels of toxic and harmful substances such as histamine. It has been reported that dietary histamine may exert harmful effects by causing oxidative stress [10], inflammatory responses [11,12] and microbial disturbances in the intestine [13]. Due to the negative effects of higher dietary histamine on the growth performance and health of eels, LQBFM could not be routinely used in eel feeds. Therefore, alleviating the damage to fish health caused by histamine in LQBFM and enhancing the physiological function of fish are the key points to solve the application limitation of LQBFM.

In recent years, some functional feed additives, including *Macleaya cordata* extract [14], grape seed proanthocyanidins [15] and compound acidifiers [16], have been individually supplemented in eel diets to improve the parameters of growth and intestinal health status of American eels (*Anguilla rostrata*). The principal biological activities of these functional additives are anti-inflammatory, antioxidant and beneficial shaping of the microflora in the intestine, respectively [17,18,19]. The beneficial effects of them on growth and health may counteract the adverse impact caused by histamine from LQBFM. This can be a practicable alternative to increase the proportion of LQBFM in the diet of eels. In addition, many studies have demonstrated that a compound of some feed additives rather than a single feed additive is more effective in improving the physiological function of fish [16,20,21,22]. The compound additives (CAs), which are composed of *Macleaya cordata* extract, grape seed proanthocyanidins and compound acidifiers, can be utilized to increase the proportion of LQBFM by playing the combined effects. Therefore, the objective of this study was to evaluate the effects of replacing WFM with LQBFM with CAs on the growth performance and intestinal health of juvenile American eels, which provided a possible way to effectively utilize LQBFM and lower the feed cost in eel culture.

## 2. Materials and Methods

### 2.1. Feeding Trial and Fish Management

Six hundred healthy juvenile American eels were acquired from Fujian Jingjiangzhiman Aquatic Technology Co., Ltd., Zhangzhou, China. The juveniles were temporarily cultured in two circular PVC tanks with a water recirculation system (800 L water, 5 L/min flow rate) to acclimate to the trial conditions for 28 days. The trial fish were hand-fed on a commercial diet twice daily (6:30 and 18:30). Before each feeding, the powder diet was mixed with 1:1.2 weight of water to form a dough and then placed on a feeding table for the eels to eat. After feeding for 30–35 min, the uneaten diet was picked up with a net, then dried and weighed to record the feed consumption.

After the acclimation period, all eels were starved for 24 h and then three hundred American eels with uniform body size (11.02 ± 0.02 g/fish) were randomly allocated to twelve tanks (400 L water). The 12 tanks were randomly assigned into 4 groups, and there were 3 replicates in each group with 25 eels per replicate. The four groups were the control group, LQBFM20+CAs group, LQBFM30+CAs group and LQBFM40+CAs group, respectively. The four groups were fed a basal diet and the basal diet with LQBFM replacing WFM at levels of 20%, 30% and 40%, respectively. Meanwhile, the diets with LQBFM were supplemented with 0.50% CAs. The CAs were a mixture of *Macleaya cordata* extract (Hunan Meikeda Biological Resource Co., Ltd., Liuyang, China), grape seed proanthocyanidins (Shandong Jinpharm Co., Ltd., Linyi, China) and compound acidifiers (Taigao Nutrition Technology Co., Ltd., Beijing, China) at the weight ratio of 1:3:30 in the diet. This ratio was based on the dietary supplementation levels of *Macleaya cordata* extract, grape seed proanthocyanidins and compound acidifiers at 100 mg/kg, 300 mg/kg and 3000 mg/kg according to the results of previous studies [14,15,16].

The formulation and amino acid concentrations of the four diets are presented in Table 1 and Table 2. The histamine levels of the four diets were 136.15 mg/kg, 252.34 mg/kg, 310.44 mg/kg and 368.53 mg/kg, respectively. The formal feeding trial lasted 70 days. The water quality parameters and fish management during the feeding trial were maintained consistently with those in the acclimation period. The water quality parameters ranged as follows: water temperature 24.5–26.5 °C, pH 6.8–7.8, ammonia nitrogen < 0.5 mg/L and dissolved oxygen > 7.0 mg/L.

### 2.2. Sample Collection

After the feeding trial, all fish were fasted for 24 h and were anesthetized with 0.1 mg/L eugenol before sampling. The eels in each tank were single-weighed for calculations of growth performance parameters. Fourteen eels in each tank were randomly captured and their whole body was wiped clean. Blood was collected from the caudal vein of fourteen eels in each tank using sterile syringes and then serum was separated by centrifugation (3000 r/min, 10 min, 4 °C) and stored at −80 °C for analysis of the serum biochemical parameters. The intact intestine samples of four eels in each tank were dissected, cleaned and stored at −80 °C for the determination of antioxidant enzymes. The midguts of four eels in each tank were sampled and rinsed with normal saline. Then, two midguts were immersed in Bouin’s solution for the preparation of intestinal sections and two midguts were immobilized in the fixative solution for electron microscopy examination. Additionally, the midguts of six eels in each tank were aseptically sampled and stored at −80 °C for profiling of the intestinal microbiota.

### 2.3. Calculations of Growth Performance

On the termination of the feeding trial, the number and weight of the eels in each tank were recorded to calculate the growth performance and feed utilization parameters using the following equations:Survival rate (SR, %) = 100 × (N_f_ − N_i_);
Weight gain rate (WGR, %) = 100 × [W_f_ − W_i_]/W_i_;
Specific growth rate (SGR, %/d) = [(Ln W_f_ − Ln W_i_) × 100]/t;
Feed intake (FI, g/fish) = 100 × FC/N_f_;
Feed conversion ratio (FCR) = 100 × FI/[W_f_ − W_i_];
Protein efficiency ratio (PER, %) = 100 × [W_f_ − W_i_]/[FC × P_d_].
where N_f_ and N_i_ were the final and initial number of eels in each tank; W_f_ and W_i_ were the final and initial weight of eels in each tank; t was the trial days; FC was the feed consumption in each tank; and P_d_ was the protein content in the trial diet.

### 2.4. Analysis of Proximate Composition and Amino Acid Concentrations

The crude protein content (Kjeldahl method by measuring nitrogen, N × 6.25), crude lipid content (ether extraction following the Soxhlet method), ash content (combustion at 550 °C for 8 h) and moisture content (drying to constant weight at 105 °C) of the diets were carried out according to the methods of AOAC [23]. The amino acid levels in the four trial diets were determined according to the description by Lu et al. [9].

### 2.5. Measurement of Serum Biochemical Parameters and Intestinal Antioxidant Parameters

The commercial assay kits were used to measure serum biochemical parameters, which included glutamic-pyruvic transaminase (GPT), glutamic-oxalacetic transaminase (GOT), acid phosphatase (ACP), alkaline phosphatase (AKP), lysozyme (LZM) and complement 3 (C3) as well as intestinal antioxidant parameters, which included total antioxidant capacity (T-AOC), superoxide dismutase (SOD), catalase (CAT), glutathione peroxidase (GSH-Px) and malondialdehyde (MDA). All the assay kits were provided by Nanjing Jiancheng Bioengineering Institute (Nanjing, China).

### 2.6. Intestinal Morphology

The sections of intestinal morphology were made according to the method used by Chen et al. [14]. Briefly, the prefixed midgut samples were gradually dehydrated with ethanol, cleared with xylene, embedded in paraffin, then cut into 4 μm slices, and finally stained using hematoxylin and eosin. The morphological parameters of villus height (VH) and muscular thickness (MT) were observed using a light microscope (BX80-JPA, Olympus, Tokyo, Japan) and morphometric analysis was carried out with Image-Pro Plus 6.0 software (Media Cybernetics, Silver Spring, MD, USA).

Intestinal scanning electron microscopy sections were prepared as follows: the prefixed midgut samples were rinsed with 0.1 M phosphoric acid buffer 3 times, dehydrated gradually with a graded series of ethyl alcohol and then dehydrated with isoamyl acetate for 15 min. The dehydrated samples were dried in a critical point dryer (K850, Quorum, Sussex, UK) then pressed onto the conductive carbon film double-sided tape and put into an ion sputtering instrument (MC1000, Hitachi, Tokyo, Japan) for gold coating. Ultrathin sections were then observed with a scanning electron microscope (SU8100, Hitachi, Tokyo, Japan).

### 2.7. Intestinal Microbiota Profiling

Given no statistical differences in growth, serum biochemical parameters and some intestinal health indicators between the LQBFM30+CAs and LQBFM40+CAs groups, intestinal samples from the control, LQBFM20+CAs and LQBFM40+CAs groups were selected for intestinal microbiota profiling. The procedures of total DNA extraction, quality detection and PCR amplification of bacterial 16S rDNA V3-V4 region were performed in accordance with the description in the study of Zheng et al. [24]. Subsequently, high-throughput sequencing was carried out on an Illumina Miseq PE300 platform. Intestinal microbiota profiling was powered by Beijing Allwegene Tech. Co., Ltd. (Beijing, China). The detailed procedure of the deep sequencing analysis was described in a previous study [25].

### 2.8. Statistical Analysis

All statistical analyses were subjected to one-way ANOVA using SPSS 22.0 software (SPSS, Chicago, IL, USA) and followed by Duncan’s multiple range test. Differences were declared significant when *p* < 0.05. Before statistical analysis, squared arcsine transformation was performed if the data were expressed as percentages. All data of the present study were expressed as means ± SD (standard deviation, *n* = 3). The alpha diversity indices of intestinal micro-organisms were analyzed using QIIME (v1.8.0) software. The linear discriminant analysis (LDA) was conducted using R-Statistical software v3.6.0 (R Statistical Corp., Vienna, Austria) with a score threshold of 3.0, *p* < 0.05.

## 3. Results

### 3.1. Growth Performance and Feed Utilization Parameters

The growth performance and feed utilization of juvenile American eels in different treatment groups are presented in Table 3. The SR values of juvenile American eels in four treatment groups were 100%. Compared to the control group, WGR, SGR, FI, FCR and PER were significantly decreased in the LQBFM30+CAs and LQBFM40+CAs groups (*p* = 0.007, *p* = 0.008, *p* < 0.001, *p* = 0.040, *p* < 0.001, respectively), while no significant differences in the parameters of growth performance and feed utilization were observed between the control group and LQBFM20+CAs group (*p* > 0.05).

### 3.2. Biochemical Parameters in Serum

As shown in Table 4, there was an increasing trend of activities of GOT and GPT with the increasing proportion of LQBFM combined with 0.50% CAs; the lowest values were both found in the LQBFM20+CAs group (*p* < 0.001). For the parameters of nonspecific immunity, no significant differences were found in the activities of ACP or AKP among all treatment groups (*p* > 0.05). The LZM activity was significantly decreased in the LQBFM30+CAs and LQBFM40+CAs groups in comparison to that of the control group (*p* < 0.001), while the difference between the control and LQBFM20+CAs groups was not significant (*p* > 0.05). The C3 level in the LQBFM30+CAs and LQBFM40+CAs groups was significantly lower than that in the control group, while the C3 level in the LQBFM20+CAs group was significantly higher than that in the control group (*p* < 0.001).

### 3.3. Antioxidant Parameters in the Intestine

As shown in Table 5, the level of T-AOC, as well as activities of SOD and GSH-Px in the LQBFM30+CAs and LQBFM40+CAs groups, were significantly decreased in comparison to those in the control and LQBFM20+CAs groups (*p* < 0.001, *p* = 0.001, *p* = 0.003, respectively) and the lowest values were all found in the LQBFM40+CAs group. The CAT activity was significantly decreased only in the LQBFM40+CAs group (*p* = 0.045), whereas no difference was observed among the other three groups (*p* > 0.05). In contrast, MDA content in the LQBFM30+CAs and LQBFM40+CAs groups was significantly higher than that in the control group and LQBFM20+CAs group (*p* < 0.001).

### 3.4. Intestinal Morphology

As shown in Figure 1 and Table 6, the VH of the intestine was significantly decreased with the increase in the proportions of LQBFM (*p* = 0.022), while no difference was observed between the control group and the LQBFM20+CAs group (*p* > 0.05). The MT of the intestine in the LQBFM30+CAs group and LQBFM40+CAs group was significantly lower than that in the control group and LQBFM20+CAs group (*p* < 0.001).

As illustrated in Figure 2, the results of the scanning electron microscopy showed that the intestinal microvilli density was sparse to different degrees with the increase in the proportions of LQBFM. The change in the microvilli density in the LQBFM20+CAs group was not obvious in comparison to that of the control group, while the microvilli in the LQBFM30+CAs and LQBFM40+CAs groups were seriously damaged.

### 3.5. Intestinal Microbiota

The alpha diversity of the intestinal microbiota of juvenile American eels is shown in Table 7. There were no significant differences in the indexes of operational taxonomic units (OTUs), Chao1 estimator (Chao1) or Simpson index (Simpson) among the three treatment groups (*p* > 0.05). The coverage rate was 100%, indicating that the majority of intestinal bacteria were identified.

As illustrated in Figure 3, the microbiota compositions of the control, LQBFM20+CAs, and LQBFM40+CAs groups were dominated by Tenericutes, Firmicutes, Proteobacteria and Spirochaetae. With the increase in the proportions of LQBFM in the diets, there was a decreasing trend in the relative abundances of Tenericutes and Proteobacteria, while there was an increasing trend in the relative abundances of Firmicutes and Spirochaetae.

As illustrated in Figure 4, the relative abundance of *Corynebacteriaceae* was higher in the control group, while the relative abundances of *Gottschalkia*, *Leucobacter*, *Gemmata*, *Iamia* and *Breznakia* were higher in the LQBFM20+CAs group and the relative abundance of *Clostridium-sensu-stricto-13* was higher in the LQBFM40+CAs group.

## 4. Discussion

In the present study, the growth performance and feed utilization in the LQBFM20+CAs group showed no significant differences compared to those in the control group. However, those growth parameters were obviously decreased in the LQBFM30+CAs and LQBFM40+CAs groups. This suggested that LQBFM combined with 0.50% CAs could replace 20% of WFM without adverse effects on growth parameters. As reported in the previous study, the replacement of WFM with BFM resulted in poor growth and feed utilization of turbot (*Scophthalmus maximus*) [26,27], possibly related to the presence of some toxic substances in BFM such as histamine. It was found that dietary histamine levels exceeding 247 mg/kg could decrease the growth performance of *A. rostrata* juveniles [13]. In the current study, the histamine levels were gradually increased with the increasing proportion of LQBFM in the diets and the histamine levels of the three trial groups were over 247 mg/kg. However, the growth performance in the LQBFM20+CAs group was similar to that in the control group; this might be caused by the supplementation of CAs. Grape seed proanthocyanidins supplemented in the diets has been found to counteract the growth inhibition of American eels induced by dietary histamine [28]. At present, studies on the alleviation of histamine-induced growth retardation of eels with *Macleaya cordata* extract and acidifiers are little reported. However, dietary supplementations of both *Macleaya cordata* extract and compound acidifiers have been confirmed to improve the growth performance of American eels effectively under conditions without exogenous stress [14,29]. Some researchers have indicated that compound feed additives could be more effective than individual feed additives in improving the physiological status and growth of fish [21,22]. It is possible that the combined effects of the CAs could eliminate the negative effects of 20% LQBFM in the diet on the growth performance of the trial fish and the alleviation ability of CAs could not be strong enough to counteract the negative effects of LQBFM at higher proportions in the diets of American eels.

Serum biochemical parameters are often used to reflect the physiological state of fish [30]. GOT and GPT are transaminases existing in liver cells. Their activities in serum will elevate after the liver suffers a certain degree of damage [31]. LZM activity as well as the C3 level are important indicators to evaluate the innate immunity of fish [9,32,33]. A previous study found that the replacement of 38% WFM with LQBFM increased the activities of GOT and GPT and decreased LZM activity in the serum of Chinese soft-shell turtles (*Pelodiscus sinensis*) [34]. It was generally considered that LQBFM contained higher levels of histamine and some other toxic substances in comparison to WFM, which could cause direct damage to the liver and disturb the serum physiological status [27]. It was found that over 200 mg/kg histamine in the diet increased the activities of GOT and GPT and decreased nonspecific immune parameters in the serum of American eels [13]. In the present study, it was noteworthy that the histamine levels of three LQBFM groups exceeded 200 mg/kg. Compared to the control group, the activities of GOT and GPT were not increased, and both LZM activity and C3 level were not lowered in the LQBFM20+CAs group. These results suggested that CAs probably mitigated the detrimental effects of 20% LQBFM on the liver function and nonspecific immunity of American eels. A similar phenomenon was also found, that grape seed proanthocyanidins supplementation in the diet decreased the activities of GOT and GPT and enhanced C3 level in the serum of American eels exposed to dietary histamine stress [15,28]. At present, little information is available about *Macleaya cordata* extract or compound acidifiers to improve the liver health status and innate immunity ability of fish, while they can exert beneficial effects on those parameters under trial conditions without stressors [14,16]. However, CAs did not produce certain beneficial effects on the health of American eels in the LQBFM30+CAs group and LQBFM40+CAs group, which could be due to the levels of histamine being too high in the diets, meaning that 0.50% CAs did not alleviate their negative effects. This should be confirmed in a future study.

T-AOC is a comprehensive indicator reflecting the ability of the organism to metabolize oxygen free radicals [35]. SOD, CAT and GSH-Px are the antioxidant enzymes against oxygen free radicals and play extremely important roles in preventing cellular oxidative damage [36]. MDA is the final product of lipid peroxidation and its level can indirectly reflect the degree of cellular oxidative [37]. It was obvious that the replacement of WFM with LQBFM increased the levels of some toxic substances like histamine in the diets. Over 200 mg/kg histamine in the diet decreased the antioxidant capacity in the intestine of juvenile American eels [13]. In the present study, the histamine level in the LQBFM20+CAs group was 252.34 mg/kg and its intestinal antioxidant potential was not decreased in comparison to that of the control group. This result was probably attributed to the protection effects of the CAs. As a natural antioxidant, grape seed proanthocyanidins was found to alleviate histamine-induced oxidative stress in American eels [15]. Although no study was conducted to evaluate the alleviation effects of *Macleaya cordata* extract or acidifiers on the oxidative stress induced by dietary histamine, those two feed additives were demonstrated to increase the antioxidant ability in the intestine of American eels under the trial condition without stressors [29,38] and the protective effects of those two feed additives could be related to the activation of the Nrf2 signaling pathway and inhibition of free radical production [39,40]. The significant decrease in the antioxidant potential in the intestine of the trial fish from the LQBFM30+CAs group and LQBFM40+CAs group indicated that the counteractive effects of CAs are probably limited to a supplementation level of 0.50%. Further investigation is required to determine whether higher levels of dietary CAs supplementation could alleviate oxidative stress caused by higher proportions of LQBFM with elevated histamine levels.

The higher values of intestinal VH, MT and microvilli density indicate that there is a higher ability for the absorption and transportation of nutrients in the intestine of fish [41,42,43]. It was found that over 200 mg/kg histamine could induce obvious damage to the structure of the intestinal mucosa and decrease the intestinal microvilli density of American eels [13]. In the present study, no statistical differences in the parameters of the intestinal morphology were observed between the LQBFM20+CAs and control groups. Those results suggested that 0.50% CAs probably mitigated the intestinal barrier damage arising from histamine in the LQBFM20+CAs group. As the feed additive was individually supplemented in the diet, grape seed proanthocyanidins was proven to protect the intestine from the morphology damage of American eels exposed to dietary histamine stress [44]. There were no reports on the effects of *Macleaya cordata* extract or compound acidifiers on the intestinal morphology of eels under dietary histamine stress. However, *Macleaya cordata* extract could ameliorate the intestinal morphology of American eels without the designed trial stressors [14,29]. *Macleaya cordata* extract could maintain the intestinal structural integrity by upregulating the expression of intestinal tight junction protein-related genes [45]. Similarly, the compound acidifiers also had positive effects on maintaining the parameters of the intestinal morphology of American eels [19], which could be ascribed to its ability to promote the proliferation of intestinal epithelial cells and decrease intestinal permeability [46,47]. Notably, the aforementioned intestinal morphological parameters in the 30% and 40% LQBFM groups with higher histamine levels were significantly poorer than those in the LQBFM20+CAs group. Dietary 0.50% CAs supplementation did not restore the intestinal damage produced by the higher levels of histamine in the diets with 30% and 40%LQBFM, which could be ameliorated by the higher supplementation level of CAs. A future study should be conducted to confirm this speculation.

Intestinal microbes play a key role in nutrient digestion, absorption and maintenance of the homeostasis of the host metabolism and immune systems [48]. Generally, the indexes of OTUs, Chao1 and Simpson obtained by alpha diversity analysis are used to reflect the species diversity and abundance of intestinal microbiota [49,50]. It was verified that the diversity or abundance of the intestinal microbiota of juvenile American eels was significantly affected by the dietary histamine levels at 350 mg/kg or 534 mg/kg [13]. In this study, the alpha diversity of intestinal microbiota showed no statistical differences among the control group, LQBFM20+CAs group and LQBFM40+CAs group. These results might be caused by 0.50% CAs supplementation in the diets. It was reported that dietary 300 mg/kg grape seed proanthocyanidins could reverse the alteration of 534 mg/kg histamine on the diversity or abundance of the intestinal microbiota of juvenile American eels [44]. At present, the effects of *Macleaya cordata* extract or compound acidifiers on the alpha diversity of intestinal microbiota were not reported under dietary histamine stress but the alpha diversity of the intestinal microbiota was affected by supplementation of those two feed additives individually without dietary stressors [14,19]. The roles of those two feed additives in regulating the intestinal microbiota under the high level of histamine in the diets of juvenile American eels should be confirmed in future research.

Tenericutes, Firmicutes, Proteobacteria and Spirochaetae were the four major phyla of bacteria in this study, which were similar to the results in the previous studies about the composition of the intestinal microbiota of American eels [14,19,44,51]. As for the changes in bacteria at the phylum level, Tenericutes might promote the growth of fish and inhibit the proliferation of pathogenic bacteria in the intestine [52]. Some bacteria in Proteobacteria could promote protein digestion in the gut and improve the metabolic state of the host [53]. Some strains of Firmicutes have been reported to be related to the reduced growth rate of some fish species [25]. Spirochaetae could be associated with hypoxic conditions impairing the intestinal health problem of fish [54]. In the previous study of juvenile American eels fed a diet with a high level of histamine, there was a decreased relative abundance of Proteobacteria and an increased relative abundance of Firmicutes in the intestine [13]. In the present study, the relative abundances of intestinal microbiota at the phyla level in the LQBFM20+CAs group and LQBFM40+CAs group were affected slightly in comparison to that in the control group. This phenomenon was probably due to the regulation effects of CAs on the intestinal microbiota to restore the changes caused by the higher levels of dietary histamine in the LQBFM20+CAs group and LQBFM40+CAs group. An increasing relative abundance of Firmicutes accompanied by a decreasing relative abundance of Proteobacteria caused by dietary 300 mg/kg grape seed proanthocyanidins were found in the intestine of juvenile American eels fed a diet with a high level of histamine [44]. Although no study was conducted on alleviating the histamine-induced intestinal microbiota disorders by *Macleaya cordata* extract or compound acidifiers, the individual supplementation of those two additives could similarly affect the relative abundances of Tenericutes, Firmicutes, Proteobacteria and Spirochaetae as shown in this study [14,38].

In the present study, the relative abundances of *Gottschalkia*, *Leucobacter*, *Gemmata*, *Iamia* and *Breznakia* were significantly increased in the LQBFM20+CAs group in comparison to those in the control group. *Gottschalkia* spp. was found only in the gut of large yellowfin but not in small and medium-sized fish, which could play an important role in growth and development [55]. The *Leucobacter* in the intestinal microbiota of tilapia was reported to be possibly associated with improved immunity [56]. The *Gemmata* was found to be higher in fish fed with probiotic-treated groups than that of the control, which might be beneficial for immune response [57]. Little information was known about the role of *Iamia* and *Breznakia* in fish [58,59]. According to the results of previous studies, there should be higher relative abundances of some pathogenic bacteria in the intestine of juvenile American eels in theory [14,19,51]. Nevertheless, there could be more probiotics in the intestine of juvenile American eels from the LQBFM20+CAs group. These results could be caused by 0.50% CAs supplementation in the diet. Similar results were also demonstrated in the study of grape seed proanthocyanidins, which increased the relative abundances of certain probiotics and lowered the relative abundances of some pathogenic bacteria in the intestine of juvenile American eels [51]. Although the specific bacteria at the genus level were not consistent in the previous study and present trial, the beneficial modification of the intestinal microbiota was confirmed. Similar changes were also observed in studies of *Macleaya cordata* extract or compound acidifiers that modulated the intestinal microbiota at the genus level in juvenile American eels under the trial conditions with no exogenous stress [14,19]. However, the exact roles of those two feed additives in regulating the intestinal bacteria at the genus level of this fish species fed a diet with a higher histamine level should be clarified in future studies. However, the relative abundance of *Clostridium-sensu-stricto-13* was increased in the LQBFM40+CAs group, which could be considered as a conditional pathogenic bacterium [60]. It suggested that 0.50% CAs could not alleviate the negative effects of a higher level of histamine in the diet with 40% LQBFM on the intestinal bacteria. Whether higher levels of dietary CAs supplementation can beneficially modify the intestinal bacteria of juvenile American eels fed diets with higher proportions of LQBFM remains to be investigated.

Taken together, the results of this study implied that LQBFM supplemented with 0.5% CAs could replace 20% of WFM in the diet without negative effects on the growth, serum biochemistry and intestinal health of American eels, which would lower the feed costs in eel culture. However, a higher proportion of replacement of WFM with LQBFM (30% or 40%) in the diet resulted in detrimental impacts on the growth performance and health status, possibly related to the hazardous effects of certain toxic substances in LQBFM. It is well-known that LQBFM has a higher content of biogenic amines. Histamine has the highest toxic effect among different biogenic amines [61] and other biogenic amines, such as cadaverine, putrescine, tyramine and monoamines, could synergistically enhance the toxicity of histamine [62,63,64]. The adverse impact of dietary histamine on the growth and intestinal health in the higher replacement groups of WFM by LQBFM might be too severe to be alleviated by CAs at the present supplementation level. Whether higher CA supplementation levels might improve the dosage of LQBFM should be confirmed in further study. Additionally, the protection effect of CAs on dietary histamine stress might provide a new way to alleviate other types of stressors in aquaculture. This specific effect needs further investigation.

## 5. Conclusions

With 0.50% CAs supplementation in the diet of juvenile American eels, LQBFM could successfully replace 20% of WFM without detrimental effects on the parameters of growth, nonspecific immunity in serum and antioxidant potential, morphology and microbiota in the intestine. However, the replacement of 30% and 40% WFM with LQBFM exerted an adverse impact on the growth and intestinal health status of this fish species. This study provides a practical application strategy to reduce the feed cost in American eel culture and it should be elucidated in further study whether higher levels or component ratios of CAs would achieve a higher replacement level of WFM with LQBFM.

## Figures and Tables

**Figure 1 animals-13-02873-f001:**
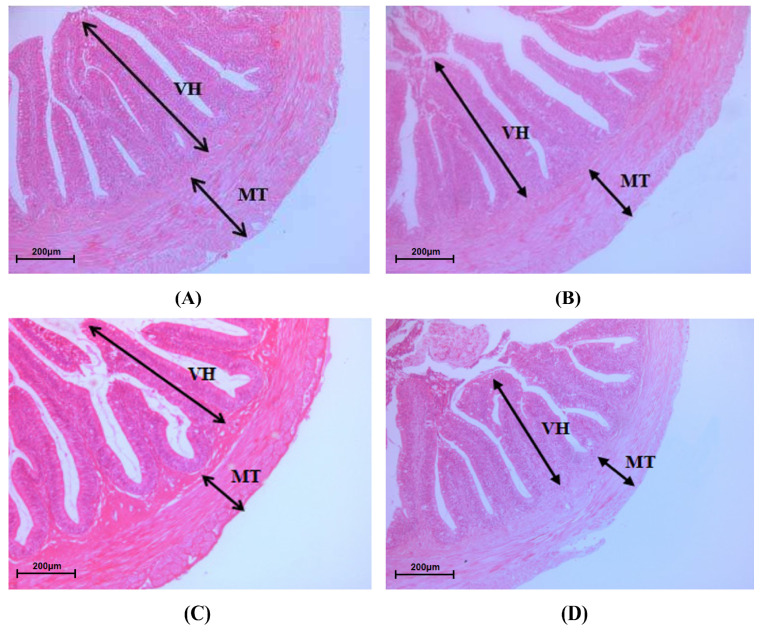
Intestinal morphology of *Anguilla rostrata* juveniles. (**A**) Control group; (**B**) LQBFM20+CAs group; (**C**) LQBFM30+CAs group; (**D**) LQBFM40+CAs group. VH: villus height; MT: muscular thickness.

**Figure 2 animals-13-02873-f002:**
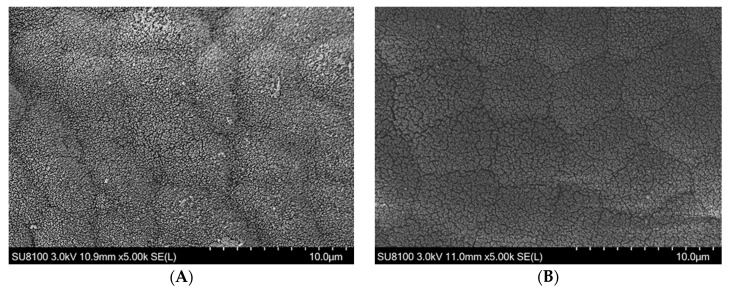
Intestinal microvilli of *Anguilla rostrata* juveniles. (**A**) Control group; (**B**) LQBFM20+CAs group; (**C**) LQBFM30+CAs group; (**D**) LQBFM40+CAs group. Instrument model: SU8100 (Hitachi Regulus 8100, Hitachi, Tokyo, Japan); working voltage: 3.0 kV; working distance (from lens to sample): 10.3–11.0 mm; magnification: 5.00 k; photo mode: SE(L).

**Figure 3 animals-13-02873-f003:**
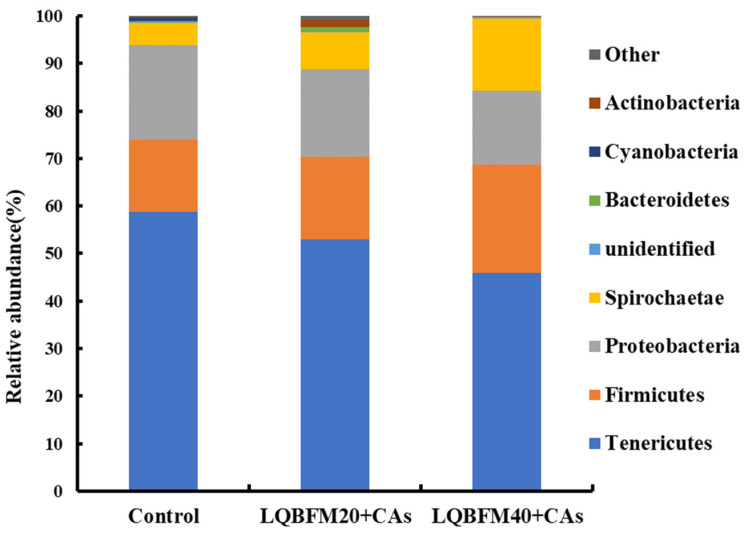
Intestinal microbiota composition at the phylum level of *Anguilla rostrata* juveniles.

**Figure 4 animals-13-02873-f004:**
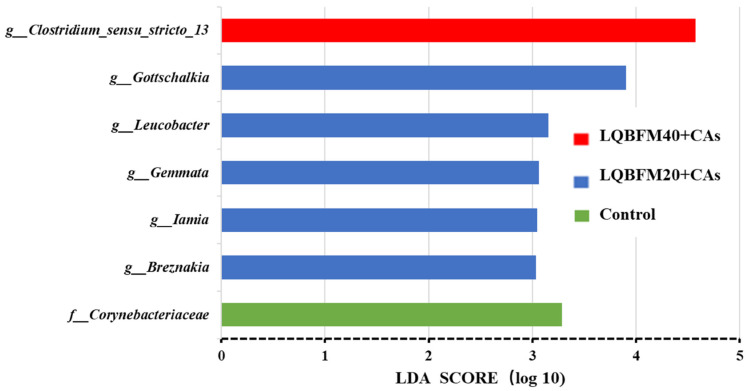
The linear discriminant analysis (LDA) in intestinal microflora at the genus level of *Anguilla rostrata* juveniles.

**Table 1 animals-13-02873-t001:** Formulation and proximate levels of the four trial diets (% dry matter).

Ingredients	Groups			
	Control	LQBFM20+CAs	LQBFM30+CAs	LQBFM40+CAs
WFM (super prime) ^a^	40.00	20.00	10.00	0.00
LQBFM ^b^	0.00	20.00	30.00	40.00
BFM (super prime) ^c^	30.00	30.00	30.00	30.00
α-starch	23.00	23.00	23.00	23.00
Extruded soybean	2.00	2.00	2.00	2.00
Brewers yeast	2.00	2.00	2.00	2.00
Fish oil	0.50	0.40	0.35	0.30
Choline chloride	0.50	0.50	0.50	0.50
Monocalcium phosphate	0.50	0.50	0.50	0.50
Vitamin premix ^d^	0.40	0.40	0.40	0.40
Mineral premix ^d^	0.60	0.60	0.60	0.60
Compound additives ^e^	0.00	0.50	0.50	0.50
Microcrystalline cellulose	0.50	0.10	0.15	0.20
Proximate analysis (% dry matter)				
Crude protein	48.49	48.35	48.72	48.23
Crude lipid	6.46	6.46	6.46	6.47
Ash	13.35	13.27	12.93	12.72

^a^ WFM: white fish meal, 67.75% crude protein, 7.44% crude lipid and 18.59% ash. It was mainly processed from cod (Alaska Coastal Airlines, Washington, DC, USA). ^b^ LQBFM: low-quality brown fish meal, 66.25% crude protein, 8.30% crude lipid and 15.78% ash. It was mainly processed from anchovy (ORIZON^®^, Orizon S.A., Coronel, Chile). ^c^ BFM: brown fish meal, 67.79% crude protein, 7.76% crude lipid and 16.29% ash. It was mainly processed from anchovy (Copeinca ASA, Lima, Peru). ^d^ Vitamin premix and mineral premix were formulated according to the previous study [9]. ^e^ Compound additives were a mixture of *Macleaya cordata* extract, grape seed proanthocyanidins and compound acidifiers in the ratio of 1:3:30.

**Table 2 animals-13-02873-t002:** Amino acid levels in the four trial diets (% dry matter).

Amino Acid	Groups			
	Control	LQBFM20+CAs	LQBFM30+CAs	LQBFM40+CAs
Threonine	2.15	2.12	2.09	2.06
Valine	2.42	2.35	2.36	2.33
Methionine	1.41	1.27	1.24	1.16
Isoleucine	2.07	1.98	2.02	1.96
Leucine	3.62	3.53	3.56	3.47
Phenylalanine	1.82	1.77	1.77	1.69
Histidine	1.36	1.48	1.55	1.61
Lysine	3.73	3.65	3.70	3.70
Arginine	2.87	2.71	2.63	2.54
Tryptophan	0.55	0.55	0.55	0.54
Essential amino acid	22.00	21.39	21.47	21.05
Aspartic acid	4.48	4.34	4.30	4.21
Glutamic acid	6.26	6.12	5.96	6.13
Glycine	3.05	2.94	2.93	2.87
Cysteine	0.50	0.44	0.42	0.40
Tyrosine	1.78	1.66	1.59	1.52
Proline	2.00	2.04	2.09	1.85
Serine	2.02	1.93	1.85	1.79
Alanine	2.98	2.98	3.06	3.05
Nonessential amino acid	23.07	22.45	22.21	21.82

**Table 3 animals-13-02873-t003:** Growth performance and feed utilization of *Anguilla rostrata* juveniles.

Items	Groups			
	Control	LQBFM20+CAs	LQBFM30+CAs	LQBFM40+CAs
IBW (g/fish)	11.02 ± 0.02	11.01 ± 0.02	11.02 ± 0.01	11.00 ± 0.02
FBW (g/fish)	24.49 ± 2.63 ^a^	25.09 ± 1.39 ^a^	21.93 ± 1.10 ^b^	20.30 ± 1.56 ^b^
SR (%)	100.00 ± 0.00	100.00 ± 0.00	100.00 ± 0.00	100.00 ± 0.00
WGR (%)	122.32 ± 11.33 ^a^	127.90 ± 11.82 ^a^	99.03 ± 9.85 ^b^	84.74 ± 11.53 ^b^
SGR (%/d)	1.14 ± 0.08 ^ab^	1.17 ± 0.08 ^a^	0.98 ± 0.07 ^bc^	0.87 ± 0.11 ^c^
FI (g/fish)	20.76 ± 0.44 ^a^	20.92 ± 0.17 ^a^	16.49 ± 0.30 ^b^	17.13 ± 0.80 ^b^
FCR	1.46 ± 0.01 ^c^	1.58 ± 0.19 ^bc^	1.76 ± 0.05 ^ab^	1.88 ± 0.22 ^a^
PER (%)	143.02 ± 0.89 ^a^	140.94 ± 1.17 ^a^	127.87 ± 5.56 ^b^	125.99 ± 0.73 ^b^

In the same row, values (means ± SD, *n* = 3) with different characters indicating significant differences (*p* < 0.05). IBW: initial body weight; FBW: final body weight; SR: survival rate; WGR: weight gain rate; SGR: specific growth rate; FI: feed intake; FCR: feed conversion ratio; PER: protein efficiency ratio.

**Table 4 animals-13-02873-t004:** Serum biochemical parameters of *Anguilla rostrata* juveniles.

Items	Groups			
	Control	LQBFM20+CAs	LQBFM30+CAs	LQBFM40+CAs
GOT (U/L)	44.09 ± 1.41 ^b^	30.25 ± 1.30 ^c^	46.62 ± 0.83 ^b^	63.01 ± 2.18 ^a^
GPT (U/L)	59.69 ± 4.06 ^c^	35.84 ± 2.60 ^d^	76.31 ± 2.87 ^b^	88.85 ± 4.21 ^a^
ACP (U/mL)	4.12 ± 0.81	5.24 ± 0.58	4.74 ± 1.46	4.98 ± 1.08
AKP (U/mL)	7.90 ± 0.37	7.31 ± 0.85	7.69 ± 0.72	7.98 ± 0.60
LZM (U/mL)	4.88 ± 0.03 ^a^	4.86 ± 0.14 ^a^	2.65 ± 0.36 ^b^	2.66 ± 0.37 ^b^
C3 (μg/mL)	132.93 ± 6.22 ^b^	155.89 ± 2.49 ^a^	98.99 ± 14.55 ^c^	66.38 ± 10.67 ^d^

In the same row, values (means ± SD, *n* = 3) with different characters indicating significant differences (*p* < 0.05). GOT: glutamic-oxaloacetic transaminase; GPT: glutamic-pyruvic transaminase; ACP: acid phosphatase; AKP: alkaline phosphatase; LZM: lysozyme; C3: complement 3.

**Table 5 animals-13-02873-t005:** Intestinal antioxidant parameters of *Anguilla rostrata* juveniles.

Items	Groups			
	Control	LQBFM20+CAs	LQBFM30+CAs	LQBFM40+CAs
T-AOC (mmol/g prot)	0.95 ± 0.01 ^a^	0.95 ± 0.02 ^a^	0.55 ± 0.04 ^b^	0.34 ± 0.06 ^c^
SOD (U/mg prot)	63.03 ± 5.05 ^a^	58.26 ± 7.75 ^a^	46.51 ± 1.77 ^b^	33.36 ± 7.20 ^c^
CAT (U/mg prot)	1.49 ± 0.22 ^a^	1.69 ± 0.05 ^a^	1.44 ± 0.04 ^a^	1.27 ± 0.16 ^b^
GSH-Px (U/mg prot)	17.57 ± 0.78 ^a^	16.55 ± 1.14 ^ab^	14.79 ± 1.43 ^b^	12.23 ± 1.26 ^c^
MDA (nmol/mg prot)	0.53 ± 0.02 ^c^	0.54 ± 0.02 ^c^	1.55 ± 0.15 ^b^	2.63 ± 0.22 ^a^

In the same row, values (means ± SD, *n* = 3) with different characters indicating significant differences (*p* < 0.05). T-AOC: total antioxidant capacity; SOD: superoxide dismutase; CAT: catalase; GSH-PX: glutathione peroxidase; MDA: malondialdehyde.

**Table 6 animals-13-02873-t006:** The VH and MT in the intestine of *Anguilla rostrata* juveniles.

Items	Groups			
	Control	LQBFM20+CAs	LQBFM30+CAs	LQBFM40+CAs
VH (µm)	530.11 ± 45.84 ^a^	524.71 ± 53.71 ^a^	472.99 ± 46.74 ^ab^	429.33 ± 23.75 ^b^
MT (µm)	143.66 ± 13.78 ^a^	159.44 ± 10.74 ^a^	120.23 ± 11.53 ^b^	101.11 ± 13.49 ^b^

In the same row, values (means ± SD, *n* = 3) with different characters indicating significant differences (*p* < 0.05). VH: villus height; MT: muscular thickness.

**Table 7 animals-13-02873-t007:** The alpha diversity of intestinal microbiota of *Anguilla rostrata* juveniles.

Items	Groups		
	Control	LQBFM20+CAs	LQBFM40+CAs
OTUs	66.66 ± 22.87	83.66 ± 42.85	62.50 ± 25.78
Chao 1	129.09 ± 68.41	109.87 ± 37.69	79.04 ± 34.26
Simpson	0.59 ± 0.12	0.63 ± 0.17	0.58 ± 0.15
Coverage rate (%)	100.00	100.00	100.00

In the same row, values (means ± SD, *n* = 3) with no character indicating no significant difference (*p* > 0.05). OTUs: operational taxonomic units; Chao1: Chao1 estimator; Simpson: Simpson index.

## Data Availability

All data generated for this study are available from the corresponding author upon reasonable request.

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
