# Peer review of "Effects of White Fish Meal Replaced by Low-Quality Brown Fish Meal with Compound Additives on Growth Performance and Intestinal Health of Juvenile American Eel (Anguilla rostrata)"

_animals, 2023, doi:10.3390/ani13182873_

Round 1

Reviewer 1 Report

The study explores an important aspect of aquaculture nutrition, and your findings have the potential to contribute significantly to the field. We appreciate the effort you have put into conducting this 70-day feeding trial and investigating the impact of replacing white fish meal (WFM) with low-quality brown fish meal (LQBFM) alongside compound additives (CAs) on the growth and intestinal health of juvenile American eels.

After reviewing your work, I have several comments and suggestions to help you enhance the quality and impact of your research:

1. Strengthen the Introduction: Provide more context on the significance of WFM in juvenile American eel diets and the reasons for exploring LQBFM as an alternative. Clearly state the research objectives and justify the need for investigating the effects of CAs supplementation on the diet.

2. Detailed Methods: Include more comprehensive information about the experimental design, including the number of replicates per treatment group and environmental conditions during the feeding trial. Provide clear details on the selection and preparation of the compound additives (CAs) and refer to relevant literature supporting their use in fish nutrition.

3. Elaborate Results and Discussion: In the Results section, consider including specific values or effect sizes for the reported statistical differences. In the Discussion, interpret the results in-depth, addressing the reasons behind the observed changes in growth performance, feed utilization, and intestinal health parameters. Compare and contrast your findings with existing literature to provide a more robust analysis.

4. Address Limitations: Acknowledge any limitations of the study and discuss potential confounding factors that may have influenced the results. Suggest directions for future research to address these limitations and further explore the topic.

5. Ethical Considerations: Ensure you mention the ethical approval obtained for the study and compliance with animal welfare guidelines, as it is essential to consider the ethical implications of using animals in research.

6. Revise Abstract and Conclusion: Restructure the abstract to adhere to the standard structure of Background, Methods, Results, and Conclusion. In the conclusion, summarize the key findings and emphasize the practical applications and benefits of your research for the aquaculture industry.

7. Language and Clarity: Thoroughly proofread and edit the manuscript to improve clarity, grammar, and sentence structure. Ambiguous sentences should be rephrased to enhance understanding.

8. Illustrations and Tables: Ensure that any figures or tables used are clear, labeled appropriately, and support the text effectively. Visual aids can significantly enhance the presentation of your findings.

9.Practical Implications: Emphasize the practical implications of your findings for the aquaculture industry and fish nutrition practices. Discuss the potential benefits and challenges of using LQBFM with CAs supplementation in eel diets. 

Overall, the quality of English language in the manuscript is good, but there are some areas that could be improved to enhance clarity and readability. Here are some specific comments:

1. Sentence Structure: The majority of sentences are well-structured and convey the intended meaning effectively. However, in some cases, the sentences are long and complex, which may make them harder to understand. Consider breaking down lengthy sentences into shorter, more concise ones to improve readability.

2. Word Choice: The choice of words and vocabulary used is appropriate for a scientific research article. However, be cautious of using uncommon or technical terms without proper explanation, as it may hinder understanding for non-specialist readers.

3. Grammar and Tenses: There are a few instances of incorrect grammar and inconsistent use of tenses. Proofreading the manuscript carefully can help identify and correct these issues.

4. Abbreviations: Ensure that all abbreviations are defined upon first mention to avoid confusion for readers.

5. Clarity: Some sentences and paragraphs could be rephrased to enhance clarity and avoid ambiguity. Always aim to convey information as clearly and precisely as possible.

6. Missing Information: In some sections, crucial information or details are missing, making it challenging for readers to fully grasp the context. Provide more information on the experimental design, methods, and results to offer a comprehensive understanding of the study.

7. Abstract Structure: The abstract should be revised to follow the standard structure of Background, Methods, Results, and Conclusion. This will help readers quickly grasp the main points of the research.

8. Cohesion and Flow: Ensure that each section of the manuscript flows logically and coherently. Use appropriate transition words to connect ideas and create a smooth reading experience.

9. Ethical Considerations: Mention the ethical approval obtained for the study and compliance with animal welfare guidelines explicitly, as it is an essential aspect of scientific research involving animals.

10. Consistency: Maintain consistency in terminology and formatting throughout the manuscript.

Paying attention to these suggestions will help you present your research more effectively and increase its impact on the readers. Overall, the work is promising, and with some refinement, it has the potential for publication.

Author Response

The study explores an important aspect of aquaculture nutrition, and your findings have the potential to contribute significantly to the field. We appreciate the effort you have put into conducting this 70-day feeding trial and investigating the impact of replacing white fish meal (WFM) with low-quality brown fish meal (LQBFM) alongside compound additives (CAs) on the growth and intestinal health of juvenile American eels. After reviewing your work, I have several comments and suggestions to help you enhance the quality and impact of your research: 1. Strengthen the Introduction: Provide more context on the significance of WFM in juvenile American eel diets and the reasons for exploring LQBFM as an alternative. Clearly state the research objectives and justify the need for investigating the effects of CAs supplementation on the diet. Re: Thanks a lot. We strengthen the Introduction section as suggested to make it clearer and more readable, please see the revised manuscript. 2. Detailed Methods: Include more comprehensive information about the experimental design, including the number of replicates per treatment group and environmental conditions during the feeding trial. Provide clear details on the selection and preparation of the compound additives (CAs) and refer to relevant literature supporting their use in fish nutrition. Re: The details information about the number of replicates per treatment group, and environmental conditions during the feeding trial were mentioned in the manuscript. The CAs selection and preparation were based on the previous reports about American eel of references, and we added related description in the revised manuscript. 3. Elaborate Results and Discussion: In the Results section, consider including specific values or effect sizes for the reported statistical differences. In the Discussion, interpret the results in-depth, addressing the reasons behind the observed changes in growth performance, feed utilization, and intestinal health parameters. Compare and contrast your findings with existing literature to provide a more robust analysis. Re: Many thanks for your advice, we added effect sizes for the reported statistical differences in the Results section. In Discussion section, we added some discussion of potential confounding factors that might affect the results and compared some indicators with the available literature, please see the revised manuscript. 4. Address Limitations: Acknowledge any limitations of the study and discuss potential confounding factors that may have influenced the results. Suggest directions for future research to address these limitations and further explore the topic. Re: Thanks a lot. We added some discussion of potential confounding factors that might affect the results and provided directions for future research. Please see the revised manuscript. 5. Ethical Considerations: Ensure you mention the ethical approval obtained for the study and compliance with animal welfare guidelines, as it is essential to consider the ethical implications of using animals in research. Re: Yes, this study was conducted according to the guidelines of the Declaration of Helsinki and was approved by the Animal Care Advisory Committee of Jimei University (Approval No. 2020-0906-005), we added this information in the revised manuscript. 6. Revise Abstract and Conclusion: Restructure the abstract to adhere to the standard structure of Background, Methods, Results, and Conclusion. In the conclusion, summarize the key findings and emphasize the practical applications and benefits of your research for the aquaculture industry. Re: Thanks a lot. We modified the Abstract and Conclusion as suggested, please see the revised manuscript. 7. Language and Clarity: Thoroughly proofread and edit the manuscript to improve clarity, grammar, and sentence structure. Ambiguous sentences should be rephrased to enhance understanding. Re: We proofread the manuscript thoroughly and revised some sentences to improve clarity, grammar and avoid ambiguity. Please see the revised manuscript. 8. Illustrations and Tables: Ensure that any figures or tables used are clear, labeled appropriately, and support the text effectively. Visual aids can significantly enhance the presentation of your findings. Re: Thanks for your suggestion, we ensured all figures and tables were checked and labeled appropriately to effectively support the text. 9. Practical Implications: Emphasize the practical implications of your findings for the aquaculture industry and fish nutrition practices. Discuss the potential benefits and challenges of using LQBFM with CAs supplementation in eel diets. Re: In Discussion, we summarized the key findings and emphasized the potential benefits and challenges of using LQBFM with CAs supplementation in eel diets. Please see the revised manuscript. Comments on the quality of English language. Overall, the quality of English language in the manuscript is good, but there are some areas that could be improved to enhance clarity and readability. Here are some specific comments: 1. Sentence Structure: The majority of sentences are well-structured and convey the intended meaning effectively. However, in some cases, the sentences are long and complex, which may make them harder to understand. Consider breaking down lengthy sentences into shorter, more concise ones to improve readability. Re: In order to improve readability, we checked the manuscript thoroughly and broke down some lengthy sentences into shorter. 2. Word Choice: The choice of words and vocabulary used is appropriate for a scientific research article. However, be cautious of using uncommon or technical terms without proper explanation, as it may hinder understanding for non-specialist readers. Re: Thanks a lot. We proofread the manuscript carefully and avoid using uncommon terms. 3. Grammar and Tenses: There are a few instances of incorrect grammar and inconsistent use of tenses. Proofreading the manuscript carefully can help identify and correct these issues. Re: Many thanks. We checked the manuscript and revised the incorrect grammar and inconsistent use of tenses. Please see the revised manuscript. 4. Abbreviations: Ensure that all abbreviations are defined upon first mention to avoid confusion for readers. Re: Thanks for your suggestion, we ensured all abbreviations were defined upon first mention, please see the revised manuscript. 5. Clarity: Some sentences and paragraphs could be rephrased to enhance clarity and avoid ambiguity. Always aim to convey information as clearly and precisely as possible. Re: Thanks a lot. We rephrased some sentences to enhance clarity and avoid ambiguity. 6. Missing Information: In some sections, crucial information or details are missing, making it challenging for readers to fully grasp the context. Provide more information on the experimental design, methods, and results to offer a comprehensive understanding of the study. Re: We added some crucial or details information about experimental design, methods, and results as suggested, please see the revised manuscript. 7. Abstract Structure: The abstract should be revised to follow the standard structure of Background, Methods, Results, and Conclusion. This will help readers quickly grasp the main points of the research. Re: Many thanks for your advice, we modified the abstract following the standard structure of Background, Methods, Results, and Conclusion. 8. Cohesion and Flow: Ensure that each section of the manuscript flows logically and coherently. Use appropriate transition words to connect ideas and create a smooth reading experience. Re: Thanks a lot. We used some appropriate transition words to connect ideas and create a smooth reading experience in the revised manuscript. 9. Ethical Considerations: Mention the ethical approval obtained for the study and compliance with animal welfare guidelines explicitly, as it is an essential aspect of scientific research involving animals. Re: All procedures involving fish were conducted according to the guidelines of the Declaration of Helsinki and was permitted by the Animal Care Advisory Committee of Jimei University (Approval No. 2020-0906-005), please see the revised manuscript. 10. Consistency: Maintain consistency in terminology and formatting throughout the manuscript. Re: We modified the manuscript carefully to maintain consistency in terminology and formatting, please see the revised manuscript.

Reviewer 2 Report

Comments to the Authors

General comments:

This study investigated the effects of low-quality brown fish meal with compound additives to replace the high-quality white fish meal on growth performance and intestinal health of juvenile American eels (Anguilla rostrata), and the results implied that low-quality brown fish meal combined with compound additives could successfully replace 20% white fish meal without adversely affecting the intestinal health status of this fish species.

The manuscript could be considered for publication after being minor revisions. Some information needs to be provided as follows.

1. In the section of Materials and Methods, it is necessary to provide the raw fish species, processing method and detailed nutrient composition of the three fish meals (white fish meal, low-quality brown fish meal and brown fish meal). 

2. In the section of Discussion, the authors should explain in details the possible reasons for the results obtained from the three fish meals.

Author Response

This study investigated the effects of low-quality brown fish meal with compound additives to replace the high-quality white fish meal on growth performance and intestinal health of juvenile American eels (Anguilla rostrata), and the results implied that low-quality brown fish meal combined with compound additives could successfully replace 20% white fish meal without adversely affecting the intestinal health status of this fish species. The manuscript could be considered for publication after being minor revisions. Some information needs to be provided as follows. 1. In the section of Materials and Methods, it is necessary to provide the raw fish species, processing method and detailed nutrient composition of the three fish meals (white fish meal, low-quality brown fish meal and brown fish meal). Re: Many thanks for your suggestion, we added some information about the raw fish species and detailed nutrient composition of the three fish meals. Please see the revised manuscript. 2. In the section of Discussion, the authors should explain in details the possible reasons for the results obtained from the three fish meals. Re: Actually, although white fish meal (Super prime, WFM), red fish meal (Super prime, BFM), and low-quality red fish meal (LQBFM) were used in the feed formula of this study, BFM was not a variable. The results obtained in this study were mainly related to the differences between WFM and LQBFM, we added some explanation of potential confounding factors that might affect the results. Please see the revised manuscript.

Reviewer 3 Report

I believe that the work "Effects of White Fish Meal Replaced by Low-quality Brown 2 Fish Meal with Compound Additives Supplementation on 3 Growth Performance and Intestinal Health of Juvenile Ameri- 4 can Eels (Anguilla rostrata) " has a high commercial and environmental appeal, but I believe that it could have improved wording in order to be more succinct and objective. Despite a whole methodology and analysis, the conclusion is very comprehensive, not specifying the points found and not making it clear which EXACT points to be studied.

Author Response

I believe that the work "Effects of White Fish Meal Replaced by Low-quality Brown Fish Meal with Compound Additives Supplementation on Growth Performance and Intestinal Health of Juvenile American Eels (Anguilla rostrata) " has a high commercial and environmental appeal, but I believe that it could have improved wording in order to be more succinct and objective. Despite a whole methodology and analysis, the conclusion is very comprehensive, not specifying the points found and not making it clear which EXACT points to be studied.

Re: Many thanks for your suggestion. We checked the manuscript carefully and revised wording to enhance clarity and readability. And we modified the Conclusion section as you suggested, please see the revised manuscript.